# Sequential Application of Calcium Phosphate and ε-Polylysine Show Antibacterial and Dentin Tubule Occluding Effects In Vitro

**DOI:** 10.3390/ijms221910681

**Published:** 2021-10-01

**Authors:** Shinechimeg Dima, Hsiao-Ting Huang, Ikki Watanabe, Yu-Hua Pan, Yin-Yin Lee, Wei-Jen Chang, Nai-Chia Teng

**Affiliations:** 1School of Dentistry, College of Oral Medicine, Taipei Medical University, 250 Wu-Hsing Street, Taipei 110301, Taiwan; shinechimeg.dima@gmail.come (S.D.); shalom.dc@msa.hinet.net (Y.-H.P.); innate19@hotmail.com (Y.-Y.L.); cweijen1@tmu.edu.tw (W.-J.C.); 2Department of Pediatric Dentistry, Boston University, Henry M. Goldman School of Dental Medicine, 635 Albany Street, Boston, MA 02118, USA; 3Department of Dentistry, Taipei Medical University Hospital, 250 Wu-Hsing Street, Taipei 110301, Taiwan; ariellehuang725@gmail.com; 4Gerodontology and Oral Rehabilitation, Graduate School of Medical and Dental Sciences, Tokyo Medical and Dental University, Tokyo 113-8510, Japan; ikki.ore@tmd.ac.jp; 5Department of Dentistry, Chang Gung Memorial Hospital, Taipei 106, Taiwan; 6Graduate Institute of Dental & Craniofacial Science, Chang Gung University, Taoyuan 333, Taiwan; 7School of Dentistry, College of Medicine, China Medical University, Taichung 404, Taiwan; 8Department of Dentistry, Shuang Ho Hospital, New Taipei 235041, Taiwan

**Keywords:** antimicrobial, polylysine, calcium phosphate precipitation, dentin tubule occlusion, dentin hypersensitivity

## Abstract

In this study, ε-polylysine and calcium phosphate precipitation (CPP) methods were employed to induce antibacterial effects and dentin tubule occlusion. Antibacterial effects of ε-polylysine were evaluated with broth dilution assay against *P. gingivalis.* CPP solution from MCPM, DCPD, and TTCP was prepared. Four concentrations of ε-polylysine(ε-PL) solutions (0.125%, 0.25%, 0.5%, 1%) were prepared. Dentin discs were prepared from recently extracted human third molars. Dentin discs were incubated with *P. gingivalis* (ATCC 33277) bacterial suspension (ca. 10^5^ bacteria) containing Brain Heart Infusion medium supplemented with 0.1 g/mL Vitamin K, 0.5 mg/mL hemin, 0.4 g/mL L-cysteine in anaerobic jars (37 °C) for 7 days to allow for biofilm formation. *P. g*–infected dentin specimens were randomly divided into four groups: CPP + 0.125% ε-PL, CPP + 0.25% ε-PL, CPP + 0.5% ε-PL, CPP + 1% ε-PL. On each dentin specimen, CPP solution was applied followed by polylysine solution with microbrush and immersed in artificial saliva. Precipitate formation, antibacterial effects, and occlusion of dentinal tubules were characterized in vitro over up to 72 h using scanning electron microscopy. ε-PL showed 34.97% to 61.19% growth inhibition levels against *P. gingivalis* (*P. g*) after 24 h of incubation. On *P. g*-infected dentin specimens, DCPD + 0.25% ε-PL, and DCPD + 0.5% ε-PL groups showed complete bacterial inhibition and 78.6% and 98.1% dentin tubule occlusion, respectively (*p* < 0.001). The longitudinal analysis on fractured dentin samples in DCPD and TTCP groups revealed deeply penetrated hydroxyapatite-like crystal formations in dentinal tubules after 72 h of incubation in artificial saliva.

## 1. Introduction

Oral biofilm is a complex microbial community growing on solid surfaces of the tooth such as enamel, root surface, and implant. The classes of antimicrobial agents used to control the oral biofilm include bisbiguanide (chlorhexidine), enzymes, essential oils, metal ions, natural molecules (plant extracts), phenols (triclosan), quaternary ammonium compounds, and surfactants [1]. The most efficient procedure for periodontitis treatment is mechanical plaque removal; however, using chemical substances in addition to mechanical cleaning has proven to be beneficial in decreasing biofilm formation. Several antimicrobials such as azithromycin [2], minocycline [3], tetracycline, metronidazole, and chlorhexidine (CHX) [4] showed better improvements in periodontal health when locally delivered in conjunction with scaling and root planning compared to scaling and root planning alone. While adjunctive locally delivered antimicrobials yield pocket depth reductions, side effects such as damage to the gastrointestinal microbiome [5] and the development of bacterial resistance to such antimicrobial agents have been reported [6,7,8]. CHX is a commonly used antibacterial agent for the inhibition of oral biofilm formation. However, previous studies have addressed the issue of antibiotic resistance [9,10] and severe hypersensitivity reactions [11,12] to chlorhexidine. Bacterial resistance and the demand for safe products has driven the need for the development of new therapies. Antibacterial peptides have been proposed as potential new approaches for prevention of dental caries [13]. Polylysine is cationic, naturally occurring polypeptide that is produced as extracellular material by *Streptomyces albulus* [14] which has been used as safe food preservative. Polylysine has broad spectrum antimicrobial activity and little resistance to bacteria, thus, its applications in biomedicine has been increasing [15].

Dentin hypersensitivity (DH) is one of the common dental diseases in adults. DH is defined as short, sharp pain arising from response to stimuli typically thermal, evaporative, tactile, osmotic, or chemical and which cannot be ascribed to any other form of dental defect or pathology [16]. The prevalence of dentin hypersensitivity ranges from 2.8% to 74%. The most accepted theory on DH, hydrodynamic theory states that stimuli applied to dentin tubules result in movement of dentinal fluid, which then stimulates nervous processes in the more pulpal areas of the dentin and/or nerves in the dental pulp itself, resulting in pain impulse transmission. Reducing the functional radius of the tubule by partially occluding the tubule orifice should greatly reduce fluid flow and, therefore, dentin sensitivity. Agents such as strontium, oxalates, sodium fluoride, calcium carbonate, bioactive glass nanoparticles, and laser are reportedly effective in occluding dentin tubules. These materials form precipitates on dentin surface which occlude dentinal tubules; however, the crystals dissolve in saliva which subsequently undermines the efficiency of those treatments [17]. 

The prevalence of DH is found to be much higher in patients with periodontal conditions. One of the major etiological agents contributing to the adult chronic periodontal disease is *Porphyromonas gingivalis*, an anaerobic, gram negative, rod bacteria [18]. *P. gingivalis* appears mainly as micro-colonies in the top layer of subgingival biofilm localization [19]. DH occurs in approximately half [20] to 98% [21] of the periodontal patients following subgingival scaling and root planning. Subgingival debridement aims to manually reduce the bacterial plaque; however, the manual debridement leaves the root dentinal tubules open which contributes for the sensitivity later. On the other hand, root surfaces when kept free from plaque become highly mineralized and display mineral depositions [22]. Therefore, keeping the dentin surface plaque free and occluding patent dentinal tubules is a crucial for reducing dentinal tubule permeability, and subsequently improving the hypersensitivity symptoms. 

The challenge is to develop material which can penetrate deeper into the tubules for long term effects. The calcium phosphate precipitation (CPP) method has exhibited potential value of occlusion of dentinal tubules [23,24]. This method aims to occlude dentinal tubules with calcium phosphate crystals which is biomimetic considering it is the main component of dentin (Figure 1). Therefore, it has been our objective in this study to investigate calcium phosphate precipitation (CPP) method consisting of two stage procedure, calcium compounds followed by natural polypeptide ε-polylysine to create effective biomimetic barriers that can show antibacterial effects against periodontopathic bacteria and precipitate deeper into the dentinal tubules. This study aims to test the null hypothesis that the application of calcium phosphate and ε-polylysine can effectively show bactericidal effect against periodontal pathogen, *P. gingivalis*, and occlude the dentinal tubules. 

## 2. Results

### 2.1. Antibacterial Efficiency of ε-Polylysine against P. gingivalis

Figure 2 shows the inhibitory effect of ε-polylysine on the *P. gingivalis* growth. All four concentrations of ε-polylysine achieved 24.31% to 35.56% growth inhibition level against *P. gingivalis* after 12 h of incubation. The growth inhibition levels steadily increased up to 24 h. At 24 h of incubation, the highest growth inhibition was observed in 0.125% ε-polylysine with 61.19%. The other three groups of ε-polylysine achieved 34.97% to 54.66% growth inhibition levels against *P. gingivalis* after 24 h incubation.

### 2.2. Characterization of Calcium Phosphate and ε-Polylysine Mixtures

The crystal phase and size of calcium phosphate and ε-polylysine mixtures were analyzed using the XRD patterns. Figure 3 shows the XRD patterns for the DCPD and MCPM mixed with different concentrations of ε-polylysine, respectively.

The MCPM when mixed with different concentrations of ε-polylysine showed precipitation of different calcium phosphate compounds, including MCPM, β-TCP, CaCO_3_, and hydroxyapatite. The XRD analysis of DCPD + 0.125% ε-PL showed the precipitation of DCPD and DCPD + 0.5% ε-PL showed the amorphous crystallization of DCPD. The other groups of DCPD showed the amorphous crystallization when mixed with ε-polylysine. The TTCP and ε-polylysine mixtures did not produce precipitates large enough to be detected by the XRD.

### 2.3. Antibacterial and Dentin Tubule Occlusion with Sequential Application of CPP and ε-Polylysine on P. g—Infected Dentin Surface

Figure 4 presents the SEM images of the control dentin discs. The 7-day biofilm dentin specimens were covered with thick *P. gingivalis* biofilm.

The SEM results following CPP and ε-polylysine application on *P. g*-infected dentin specimens are shown in Figure 5. Successful bacterial inhibition and partially and fully occluded dentin tubules were observed following CPP and ε-polylysine application on the *P. g*-infected dentin discs.

Complete bacterial inhibition and the highest dentin tubule occlusion rate was observed in DCPD + 0.5% ε-PL group with 98.19 ± 3.31% occlusion rate compared to other DCPD groups on *P. g*-infected dentin specimens (Figure 6a). DCPD + 0.125% and DCPD + 0.25% ε-PL groups showed bacterial inhibition and occlusion degree of 72.52 ± 22.18% and 78.69 ± 12.8%, respectively. In MCPM + ε-PL treatment groups, the bacterial inhibition was minimal and a great quantity of live adherent bacterial cells on dentin surface was observed (Figure 5). Following MCPM and ε-polylysine application, the calcium phosphate precipitates with visible dead bacterial cells were formed on dentin surface as early as after the application. The precipitates were larger than the tubule orifice and formed non-homogenously on the dentin surfaces. The highest tubule occlusion degree was observed in the MCPM + 0.125% e-PL group (8.33 ± 23.41%). The MCPM + 0.25% ε-PL and MCPM + 1% ε-PL groups showed the tubule occlusion degree of 7.48 ± 24.43% and 7.4 ± 23.04%, respectively. The tubule occlusion degree in MCPM + 0.5% ε-PL group was 6.69 ± 23.22%. The differences in the mean tubule occlusion degree between MCPM groups were not statistically significant (Figure 6b). In TTCP + ε-PL groups, complete bacterial inhibition and partial dentin tubule occlusion with calcium phosphate precipitates was observed (Figure 5). The highest tubule occlusion degree in TTCP + ε-PL groups was observed in TTCP + 0.25% ε-PL group (72.07 ± 12.81%) after 6 h of incubation following the application. The tubule occlusion degree in TTCP + 0.125% ε-PL and TTCP + 0.5% ε-PL groups were 62.87 ± 25.12% and 66.9 ± 27.72%, respectively. The TTCP + 1% ε-PL group showed the occlusion degree of 54.57 ± 18.28%. The differences in the mean tubule occlusion degree between four TTCP groups were not statistically significant (Figure 6c). There was no statistically significant difference between DCPD and TTCP groups in the same concentration of ε-polylysine.

### 2.4. Longitudinal Analysis of Dentinal Tubules after Calcium Phosphate Precipitation and ε-Polylysine Treatment

The length of the crystals formed in the dentinal tubules after 72 h of incubation in artificial saliva at 37 °C following the calcium phosphate precipitation and ε-polylysine combined treatments were analyzed on the SEM images of the fractured dentin discs (Figure 7).

The numbers of the chisel fractured dentine specimens with intact crystals inside the dentinal tubules that are appropriate for the statistical analysis were small. The deepest penetrated crystal length was found in DCPD + 0.25% ε-PL group after 72 h of incubation. The mean crystal length within dentinal tubules in this group was 10.76 ± 5.11 µm (*n* = 9). The crystals formed within dentinal tubules following DCPD + 0.5% ε-PL treatment and 72 h of incubation showed the mean length of 4.45 ± 1.58 µm (*n* = 9). The mean crystal length (*n* = 2) in DCPD + 0.25% ε-PL group was 5.2 ± 2.59 µm. The crystals formed after 72 h of incubation following MCPM and ε-PL treatment were adherent on the dentin surface and did not penetrate inside the dentinal tubules. The longitudinal dentinal tubule analysis in TTCP and ε-PL groups showed crystals penetrated inside dentinal tubules after 72 h of incubation. The mean crystal length in TTCP + 0.125% ε-PL group was 3.72 ± 1.32 µm. The TTCP + 0.25% ε-PL and TTCP + 0.5% ε-PL groups exhibited crystals with mean lengths of 3.51 ± 1.64 µm and 3.23 ± 1.54 µm, respectively.

## 3. Discussion

In this study, we found ε-polylysine was capable of achieving antibacterial and dentin tubule occluding dual effects when applied with calcium phosphate solutions on bacterial biofilm induced dentin surface. This effect is particularly desiring in the treatment of dentin hypersensitivity, because previous studies demonstrated that plaque should be removed to prevent the appearance and recurrence of dentin hypersensitivity. The dentinal tubules were occluded and the area of open tubules decreased in the presence of plaque removal when dentin slabs were placed in oral environment [25]. In vivo study has also demonstrated that in the plaque not controlled group the diameter of the open tubules increased over time, whereas in the plaque controlled group, the open dentinal tubules occluded with the precipitate [26].

It is desirable that treatment of dentin hypersensitivity induces natural recovery. In this study, we found hydroxyapatite (HAP)-like crystallizations within dentinal tubules in DCPD + ε-PL and TTCP + ε-PL groups after 72 h of incubations in artificial saliva. The initial effect of CPP and + ε-PL method is precipitation of the minerals onto dentin surfaces. The diffused calcium and phosphate ions to the tooth surface from supersaturated solutions form nucleation which further crystallizes by the diffusion of the ions related to the dentin and artificial saliva. In the presence of fluoride, DCPD converts into HAP (Ca_10_(PO_4_)_6_(OH)_2_). Hydroxyapatite (HAP) is the main constituent of teeth and bone, thus it very desirable as a natural occluding material. The DCPD conversion into calcium deficient HAP is proposed as follows [27]:10CaHPO_4_·2H_2_O → Ca_10_(PO_4_)_6_(OH)_2_ + 18H_2_O + 12H^+^ + 4PO_4_^3−^(1)

Previous studies have successfully experimented HAP liquid precipitate, dry sol gel powder [28], dentifrice [29], and HAP bioceramics [30] for treatment of dentin hypersensitivity. In a previous study, an attempt was made to synthesize HAP by hydrothermal means to acquire nano-particles to infiltrate to dentin tubules; however, crystallites were larger than dentin tubule diameters [31]. The calcium phosphates used in this study, DCPD and TTCP, provided precipitate particle size ideal for penetration into dentinal tubules. To ensure sustainable relief from hypersensitivity symptoms, it is desirable that desensitizing agents penetrate into tubules to a sufficient depth. This study measured the depth of tubule penetration by crystallizations in 72 h following the CPP and ε-polylysine treatment. In DCPD + ε-PL groups, we found HAP-like crystals penetration measuring up to 21 µm into the dentinal tubules. Such deep penetrations of desensitizing agents are beneficial to withstand the daily challenges in the oral cavity, including daily acid and mechanical challenges such as brushing, to ensure the long-term success of the treatment.

The results of the present study showed antibacterial effect of ε-polylysine differed when applied with different calcium phosphate compounds. According to the SEM results, the bacterial killing effect was minimal in the MCPM group compared to DCPD and TTCP groups. This may be explained with different pH of CPP solutions, calcium phosphate ratio in different CPP solutions, electrostatic interaction of ε-polylysine with calcium and phosphate ions, and precipitation rates. This study found MCPM did not provide desirable dentin tubule occluding effects when combined with ε-polylysine compared to DCPD and TTCP groups. Moreover, when applied after MCPM solution, ε-polylysine did not exhibit the same bacterial killing effect that was observed when applied after DCPD and TTCP. The calcium phosphate precipitation on the dentin surface was rapid following the MCPM + ε-PL application. The precipitate size was greater than tubule orifices, thus it prevented them from penetrating into the tubule effectively. The number of calcium and phosphate ions available in the MCPM solution is greater than that found in the DCPD and TTCP groups due to the high solubility rate of MCPM. Thus, MCPM and ε-polylysine precipitation rate is rapid to allow bacterial inhibition. Moreover, the positively charged side groups in the cationic polypeptide may electrostatically interact with the negatively charged phosphate groups in the MCPM compound (Ca/P = 0.5) which may have prevented polypeptide’s electrostatic interaction with bacterial cells, thus preventing polylysine from exhibiting its bacterial killing effects.

## 4. Materials and Methods

### 4.1. Materials

Calcium phosphate precipitate (CPP) solution was prepared by dissolving monocalcium monohydrate [Ca (H_2_PO_4_)_2_ H_2_O] (MCPM), dicalcium phosphate dihydrate [(CaHPO_4_) 2H_2_O] (DCPD) and tetracalcium phosphate [Ca_4_(PO_4_)_2_O] (TTCP) in 0.001 mol/L phosphoric acid (H_3_PO_4_) in room temperature. The ε-polylysine stock solution containing 25% ε-polylysine was obtained from Chisso Co., Tokyo, Japan. Four concentrations of ε-polylysine (ε-PL) solutions (0.125%, 0.25%, 0.5%, 1%) were prepared by diluting stock 25% ε-polylysine in double distilled water. The crystalline phases of the calcium phosphate powders mixed with different concentrations of ε-polylysine were determined using X-ray diffraction (XRD, DMX-2200, Rigaku, Tokyo, Japan).

### 4.2. Inhibitory Effect of ε-Polylysine on P. gingivalis Growth

*Porphyromonas gingivalis* ATCC332277 (*P. g*) cultures were grown anaerobically in Brain Heart Infusion (BHI) broth (Difco Laboratories Inc., Detroit, MI, USA) supplemented with 0.1 g/mL vitamin K, 0.4 g/mL L-cysteine and 0.5 mg/mL hemin at 37 °C. The antibacterial efficacy of ε-polylysine was determined using broth dilution assay. ε-Polylysine was added to the starting inoculum of bacterial suspension with 10^9^ bacterial cells to obtain four concentrations of ε-polylysine solutions (0.125%, 0.25%, 0.5%, and 1%) when diluted in BHI broth. The test solutions were incubated anaerobically at 37 °C up to 24 h. Periodically, 100 µL was collected and serial dilution was performed in PBS and inoculated on BHI with 5% sheep blood agar plates. The plates were incubated anaerobically at 10% CO_2_ for 2 days and colony forming units (CFU) were counted. The experiments were performed in triplicate. The antibacterial efficacy of ε-polylysine against *P. gingivalis* was determined by growth inhibition level. Growth inhibition level (%) was calculated by the formula; $GIL\;\left(\%\right)=\frac{Pc-Pt}{Pc}\times 100$, where *Pc* is control bacterial population and *Pt* is bacterial population in the test solution. Bacterial populations from the same time period were considered.

### 4.3. Experimental Design

To stimulate dentin hypersensitivity and evaluate the antibacterial effects of ε-polylysine on *P. gingivalis*, in vitro *P. gingivalis*, infected dentin disc models are used. The dentin specimens were obtained from the cervical area (4 m × 4 m × 1 m) of recently extracted human third molar teeth after removing the crown and root sections with diamond saw (Isomet; Buehler, IL, USA). The dentin discs were polished with 600–1000 grit silicon carbide paper to remove the cementum, soaked in 1 M acetic acid for 30 s, rinsed with ddH_2_O and autoclaved for 20 min at 121 °C. The dentin side placed up, the discs were placed in 24-well plates and 2 mL bacterial suspension containing *P. gingivalis* (10^5^ CFU/mL) was inoculated into the each well. The wells were incubated anaerobically for 7 days at 37 °C. The BHI medium was changed every two days. The *P. g*-infected dentin discs were rinsed in PBS (2×) before proceeding for a treatment.

*P. g*-infected dentin were allocated into five groups: CPP + 0.125% ε-PL, CPP + 0.25% ε-PL, CPP + 0.5% ε-PL, CPP + 1% ε-PL, negative control (untreated). On each dentin specimen CPP solution was applied by microbrush followed by ε-polylysine application with microbrush. The treated *P. g*—dentin specimens were then proceeded for surface SEM analysis. Another subgroup of *P. g*—infected dentin specimens were prepared for dentin tubule longitudinal analysis. The dentin specimens were treated with CPP and ε-polylysine applications and immersed in artificial saliva at 37 °C for 72 h. Artificial saliva was prepared from 20 mM HEPES, 1.5 mM CaCl_2_, 0.9 mM KH_2_PO_4_, 130 mM KCl, 1 mM NaN_3_ and pH was adjusted with 1 M KOH at 7.0. Artificial saliva was replenished every day (24 h). The specimens were rinsed with ddH_2_O and fractured with chisel to expose longitudinal dentinal tubules. The dentin discs were fixed, dehydrated, and sputter-coated with Au-Pd on the fractured site. Scanning electron microscopy (HITACHI su3500, Tokyo, Japan) was carried out to assess the bacterial inhibition and crystal formation on the surface and fractured sites of the dentin specimens.

### 4.4. Statistical Analysis

The mean tubule diameter in the dentin tubules was obtained by measuring 20 tubules on each disc. The tubule diameter and occluded area was measured to evaluate the change in tubule diameter before and after treatment using ImageJ software (National Institutes of Health and the Laboratory for Optical and Computational Instrumentation (LOCI, University of Wisconsin)). Data are presented as the mean ± standard deviation (SD). Statistical analyses between groups in differences in tubule occlusion percentage were calculated by ANOVA test. The *p*-values < 0.05 were set for statistical significance.

## 5. Conclusions

Calcium phosphate compounds followed by ε-polylysine application showed excellent antibacterial and dentin tubule occluding effects in vitro. Calcium phosphate precipitation method combined with natural antibacterial polypeptide showed promising results for dentin hypersensitivity treatment. Considering the limitation of this study, various oral diseases like periodontitis are strongly associated with dysbiotic microbial communities. Multispecies biofilm model should be tested in this study. Further research should include acid resistance of the occlusion crystals and long term ε-PL bacterial inhibition effect.

## Figures and Tables

**Figure 1 ijms-22-10681-f001:**
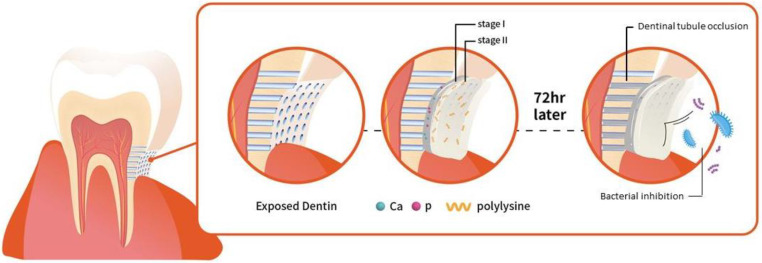
Schematic demonstration of the two steps tooth coating material inducing effective dentinal tubule occlusion and bacterial inhibition.

**Figure 2 ijms-22-10681-f002:**
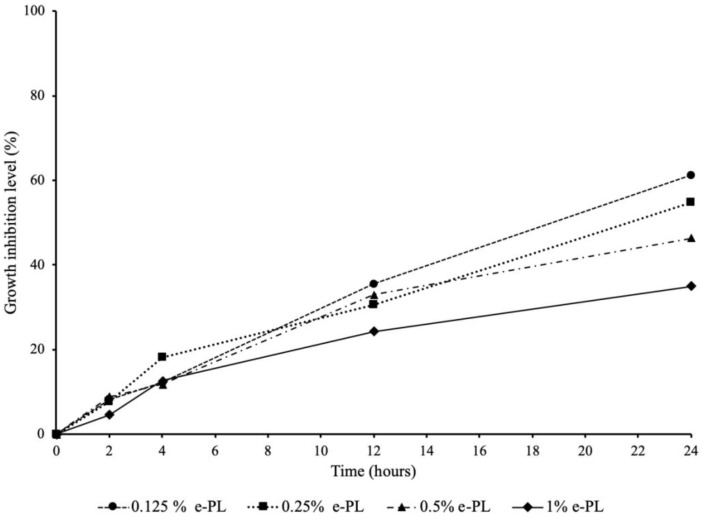
Antibacterial effect of ε-polylysine on the growth of *P. gingivalis*.

**Figure 3 ijms-22-10681-f003:**
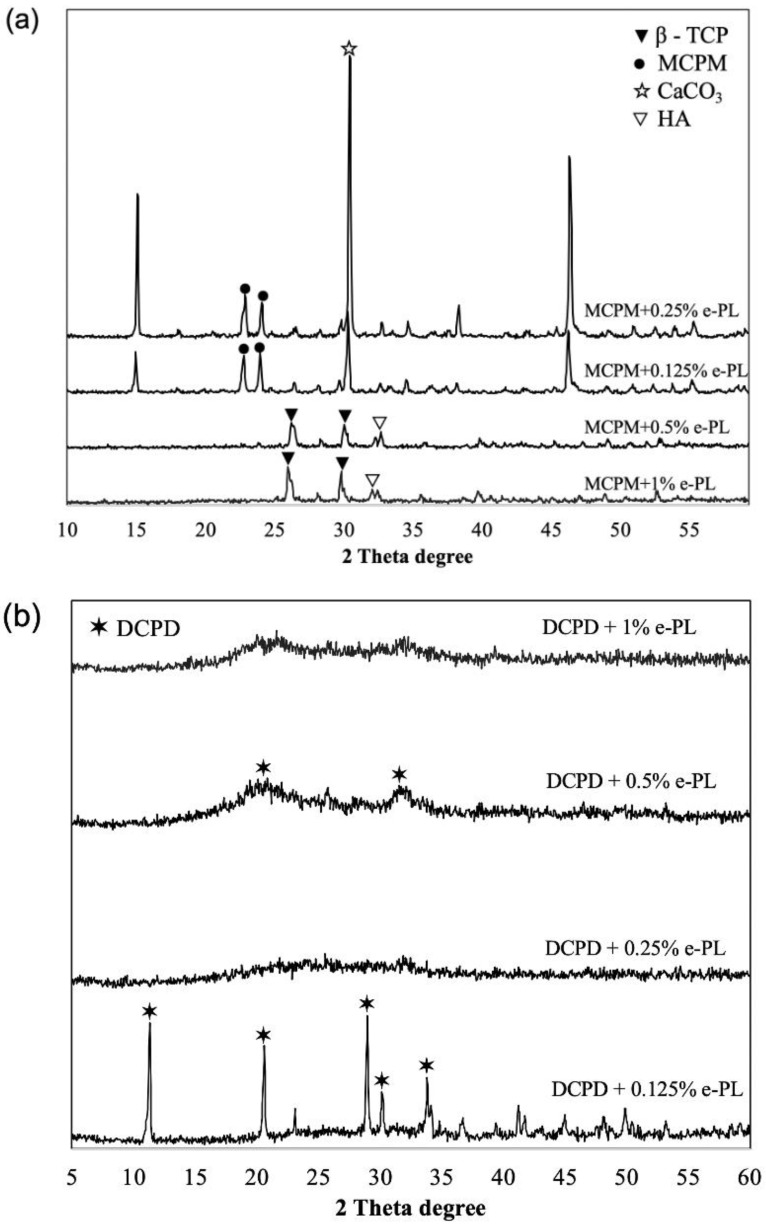
XRD patterns of MCPM (**a**) and DCPD (**b**) mixed with ε-polylysine.

**Figure 4 ijms-22-10681-f004:**
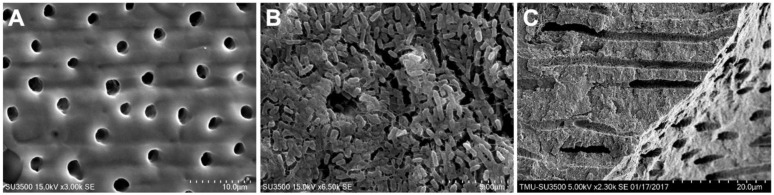
SEM images of the control dentin disc surface, (**A**) dentin disc surface with open dentin tubules; (**B**) dentin surface with 7-day *P. gingivalis* biofilm; (**C**) longitudinal dentin tubules of control dentin disc.

**Figure 5 ijms-22-10681-f005:**
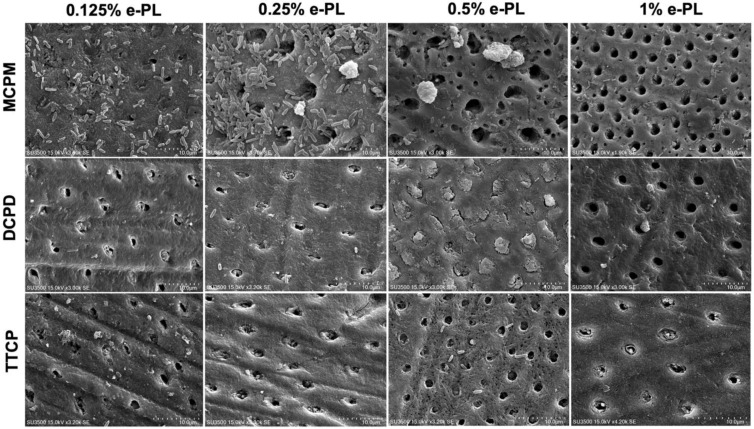
SEM images of the dentin tubule occlusion and bacterial inhibition on the *P. gingivalis* dentin surface after CPP and ε-PL application.

**Figure 6 ijms-22-10681-f006:**
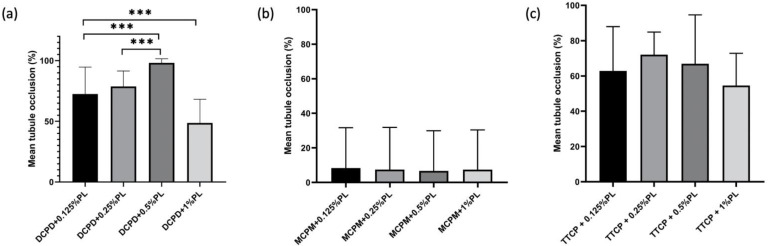
Dentin tubule occlusion rate in three groups after CPP and ε-PL application. (*n* = 10). (**a**) occlusion rate of DCPD groups on *P. g*-infected dentin specimens (**b**) occlusion rate of MCPM groups on *P. g*-infected dentin specimens (**c**) occlusion rate of TTCP groups on *P. g*-infected dentin specimens. *** *p* < 0.001.

**Figure 7 ijms-22-10681-f007:**
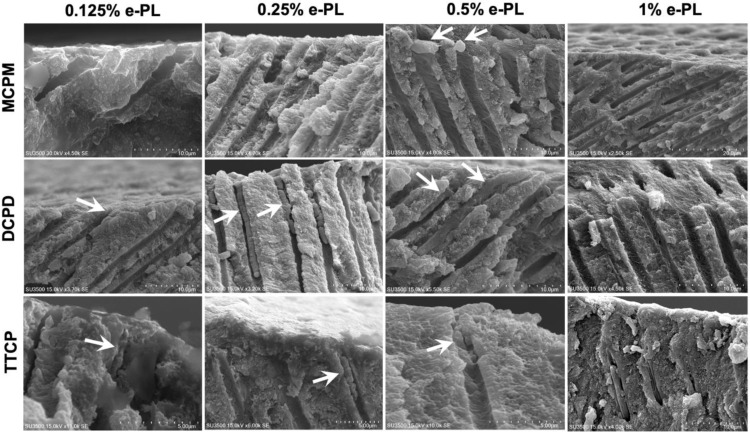
SEM images of longitudinal fractured dentin discs after 72 h of incubation in artificial saliva following CPP and ε-PL application. Arrows depict the crystallizations formed inside the dentin tubules.

## Data Availability

All data generated or analysed during this study are included in this published article.

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
