# Peer review of "Sequential Application of Calcium Phosphate and ε-Polylysine Show Antibacterial and Dentin Tubule Occluding Effects In Vitro"

_ijms, 2021, doi:10.3390/ijms221910681_

Round 1

Reviewer 1 Report

Congratulations on your work which, I found interesting.

Manuscript: Sequential application of calcium phosphate and e-polylysine show antibacterial and dentin tubule occluding effects in vitro, it is well written with an adequate structure as a scientific paper demands.

I have some minor revisions to propose to you to improve your work. Please refer to the following comments:

In Abstract: Line 27-28 is missing a verb in sentence “On each dentin specimen…”

In Material and methods: in line 144 you wrote: “the dentin specimens… were immersed for 72 hours” and then in line 147 you wrote “artificial saliva was replenished every week”. Can you explain it.

In the „Conclusions” - Please indicate a further research plan and limitations of the study

The literature is old, with many citations of works from before 2012 - please consider updating it.

Author Response

Response: Authors thank reviewer for the constructive comments on our manuscript entitled “Sequential application of calcium phosphate and Ɛ-polylysine show antibacterial and dentin tubule occluding effects in vitro” by Dima et al. The detailed comments have been addressed as follows:

Comment 1: In Abstract: Line 27-28 is missing a verb in sentence “On each dentin specimen…”

Response: We re-write the sentence as “On each dentin specimen, CPP solution was applied followed by polylysine solution with microbrush and immersed in artificial saliva.”

Comment 2: In Material and methods: in line 144 you wrote: “the dentin specimens… were immersed for 72 hours” and then in line 147 you wrote “artificial saliva was replenished every week”. Can you explain it.

Response: yes, “every week” was a typo, we make it correct as “every day (24hrs)”

Comment 3: In the „Conclusions” - Please indicate a further research plan and limitations of the study

Response: Thanks, we add this part in conclusion: “Considering the limitation of this study, various oral diseases like periodontitis is strongly associated with dysbiotic microbial communities. Multispecies biofilm model should be test in this study. Further research should include acid resistance of the occlusion crystals and long term Ɛ-polylysine bacterial inhibition effect.”

Comment 4: The literature is old, with many citations of works from before 2012 - please consider updating it.

Response: Thanks, we updated several references, especially for the recent point of view of Antibiotic resistance of periodontal pathogens.

Reviewer 2 Report

The work is interesting and well written. The article is structured correctly and all parts of the manuscript are properly compiled. I am enthusiastic and would only like to suggest a few minor corrections.

Introduction

  1. “ Agents such as strontium, oxalates, sodium fluoride, calcium carbonate, bioactive glass nanoparticles and laser are reportedly effective in occluding dentin tubules. These materials form precipitates on dentin surface which occlude dentinal tubules, however, the crystals dissolve in saliva which subsequently undermines the efficiency of those treatments.” – citations are needed

Methods

  1. How many samples were used per group?
  2. Subsection 2.4 - a large part of this fragment (lines 153-160) does not concern statistical analyzes.

Results

  1. 2 – I think that you must show standard deviations, also in 3.1. description. Where are the results of statistical analyses?
  2. In my opinion you can paste larger microphotographs.
  3. Dentin tubule occlusion rate in three groups – you should show also statistical comparison between three groups after CPP and the same PL concentration. It if important if we look at DCPD and TTCP groups.
  4. The number of the intact crystals inside the dentinal tubules – where are the statistical analyses?

Author Response

Dear reviewer:

I attached a figure in the response. The figure may not proper show here so Please see the attachment document as my response.

Thanks,
